# Coordinated Locomotion Control for a Quadruped Robot with Bionic Parallel Torso

**DOI:** 10.3390/biomimetics10050335

**Published:** 2025-05-20

**Authors:** Yaguang Zhu, Ao Cao, Zhimin He, Mengnan Zhou, Ruyue Li

**Affiliations:** 1Key Laboratory of Road Construction Technology and Equipment, Ministry of Education, Chang’an University, Xi’an 710064, China; caoao@chd.edu.cn (A.C.); hezhimin@chd.edu.cn (Z.H.); zhoumengnan@chd.edu.cn (M.Z.); liruyue@chd.edu.cn (R.L.); 2The State Key Laboratory of Mechanical Transmissions, Chongqing University, Chongqing 400044, China

**Keywords:** bionic torso, coordinated locomotion, model predictive control, variable mass center

## Abstract

This paper presents the design and control of a quadruped robot equipped with a six-degree-of-freedom (6-*DOF*) bionic active torso based on a parallel mechanism. Inspired by the compliant and flexible torsos of quadrupedal mammals, the proposed torso structure enhances locomotion performance by enabling coordinated motion between the torso and legs. A complete kinematic model of the bionic torso and the whole body of the quadruped robot is developed. To address the variation in inertial properties caused by torso motion, a model predictive control (*MPC*) strategy with a variable center of mass (*CoM*) is proposed for integrated whole-body motion control. Comparative simulations under trot gait are conducted between rigid-torso and active-torso configurations. Results show that the active torso significantly improves gait flexibility, postural stability, and locomotion efficiency. This study provides a new approach to enhancing biomimetic locomotion in quadruped robots through active torso-leg coordination.

## 1. Introduction

From a biological standpoint, a key distinction between quadrupedal robots and animals lies in the structure of the torso. Most quadrupedal mammals possess a compliant and flexible torso [1], which plays a crucial role in enhancing speed during high-speed gaits, improving locomotion efficiency, reducing muscular effort, and increasing stability in low-speed gaits. Further anatomical studies have revealed that the lumbar vertebrae in mammals are finely tuned to meet specific biomechanical demands, with the morphology of the torso being shaped by natural locomotion patterns [2]. These biological insights collectively underscore the significance of the torso in animal locomotion, emphasizing that implementing a flexible torso is imperative for quadrupedal robot design.

To date, numerous robotic designs have drawn inspiration not only from the limb structures of animals but also from their flexible torsos. For example, when a cheetah runs at high speeds, the pitching motion of its torso facilitates full extension of both forelimbs and hindlimbs while simultaneously absorbing ground reaction forces transmitted to the torso [3]. Based on structural properties and actuation mechanisms, robotic torso designs can be classified into passive and active types. Kani designed a flexible torso capable of bidirectional pitching and demonstrated that this flexibility is crucial for maintaining balance during passive hopping in quadrupedal robots [4]. Chong et al. developed a Salamandra robot with a laterally free-moving back, which improved forward, rotational, and lateral stride displacement while expanding the reachable target positions of the robot’s feet within a gait cycle [5]. Se designed the quadrupedal robot ELIRO, which incorporates a passive yaw joint in the torso, demonstrating that the torso enables smooth turning maneuvers [6]. Takuma et al. developed a quadrupedal robot with a torso exhibiting viscoelastic mechanical variations, which, despite lacking mechanical joints, achieved passive swinging [7]. To mimic biological torso structures, Ikeda introduced a passive elastic joint at the torso’s center to facilitate energy exchange and reduce energy consumption [8]. Zhang et al. showed that a compliant torso significantly enhances coordination and postural stability in robots [9]. Wang et al. experimentally confirmed that reducing the torso pitch angle is beneficial for minimizing energy expenditure during high-speed running, and passive torso joints contribute to efficient high-speed locomotion in quadrupedal robots [10].

MIT Cheetah features a flexible active torso coupled with the hind legs, enabling bionic kinematics and acting as a rear-body spring to improve energy efficiency [11]. Laika, a quadrupedal robot with a flexible active torso, incorporates a tensegrity structure that helps maintain balance when traversing obstacles such as rocks [12]. Cho et al. proposed a torso mechanism capable of rapid repeated bending and stretching, achieving a velocity of 1.0667 m/s, while the phase difference between torso and leg motion influenced locomotion patterns [13]. Kim et al. demonstrated that torso joint motion increases the forward velocity of the center of mass [14]. Kawasaki et al. developed a small quadrupedal robot, Sugoi-Neco, featuring an active torso that enables longer jumping distances [15]. Lei Jingtao et al. proposed a bionic flexible body mechanism that achieves bending by adjusting the gas pressure of artificial muscles, with a maximum bending angle of 18° recorded in experiments [16]. Qian Wei et al. designed a rigid-flexible coupled continuum bionic parallel torso capable of continuous bending in the lateral plane. Due to its underactuated properties and compliance, the torso can achieve large-amplitude flexible swinging, addressing the limitation of rigid torso constraints in traditional quadrupedal robots. However, their study did not explore coordinated motion with the legs [17]. Wang Qi et al. investigated articulated active-torso quadrupedal robots and found that coordinated torso-leg motion increases stride length and enhances running speed [18]. Zhang Chunsong et al. from Tianjin University proposed a reconfigurable quadrupedal robot, demonstrating through simulations and experiments that torso torsion effectively increases stride length [19]. Lei Jiang et al. from Zhejiang University proposed a quadruped robot with a spine, capable of performing high-dynamic activities. Their findings indicate that the flexible spine significantly impacts the robot’s motion, particularly enhancing its aerial phase performance [20].

Despite significant advancements in the research of biomimetic torso structures, most existing quadruped robot designs still suffer from limitations in degrees of freedom, often being constrained to sagittal plane motions or realized through simplified serial-link mechanisms. While such designs offer advantages in structural simplicity or compliance, they frequently compromise load-bearing capacity, directional agility, and dynamic adaptability, thereby limiting their applicability in high-performance locomotion tasks. In our previous studies [21,22,23], we proposed a fully actuated six-degree-of-freedom (6-*DOF*) biomimetic parallel torso and systematically analyzed its workspace, stiffness modulation, neuromechanical responsiveness, and damping characteristics. Comprehensive comparisons with conventional single-degree-of-freedom and passive structures demonstrated that the actively controlled parallel mechanism significantly enhances multidirectional mobility, impact mitigation capability, and coordination between the torso and limbs.

Building upon this foundation, the present study advances from structural modeling to integrated locomotion control, with a specific focus on realizing active coordination between the torso and legs in quadruped robots. A compact and efficient bionic parallel torso is developed, and a model predictive control (*MPC*) framework incorporating a variable center of mass (*CoM*) is introduced to address the inertia variations induced by torso movement. Under a trot gait, the proposed system employs gait-phase-synchronized control to replicate biologically realistic whole-body locomotion. Simulation results indicate that the method significantly improves postural stability, foot-end workspace coverage, and locomotion efficiency, thereby validating the effectiveness and advantages of active torso–leg coordination under dynamic gait conditions.

The subsequent sections of this paper are structured as follows: Section 2 introduces the structure and kinematic model of the bionic active torso, Section 3 describes the structure and overall kinematics of the quadrupedal robot with an active torso, Section 4 presents the trajectory planning for coordinated torso-leg motion, Section 5 validates the coordinated motion of the torso and legs through simulations and compares the overall motion with that of a rigid-torso robot, and Section 6 concludes the paper.

## 2. Bionic Active Torso Modeling

### 2.1. Overview of the Active Torso Model

The structure of the bionic active torso is illustrated in Figure 1. Three L-shaped brackets for securing spherical hinges are evenly distributed on the moving platform. Each L-shaped joint has two through-holes for fixation, with two spherical hinges considered as a pair. Three inclined connection brackets for securing DC motors are evenly distributed on the fixed platform, with the plane of each connection bracket forming a 60° angle with the fixed platform. Two motors are mounted in parallel on each connection bracket, considered as a motor pair. Each kinematic chain consists of a link and a rocker. The link is formed by two short links, a tension-compression force sensor, and a coaxially arranged threaded connection with a rod-end spherical hinge link. The force sensor is used to monitor the real-time force conditions of the six links in the active torso.

One end of each kinematic chain is connected to the output shaft of a DC motor, while the other end is linked to the spherical hinges on two adjacent L-shaped joints of the moving platform. Each motor output shaft can be regarded as a revolute joint, and the rotation of the rocker driven by the motor output shaft induces the movement of the kinematic chain. The simultaneous movement of the six kinematic chains results in the positional and orientational changes of the moving platform. At this stage, the moving platform and the fixed platform are supported by six kinematic chains, forming a complete six-degree-of-freedom (6-*DOF*) bionic active torso.

The designed bionic active torso is connected to the front and rear sections of the quadrupedal robot, with the fixed platform at the front and the moving platform at the rear, as shown in Figure 2. It is well known that the torso in animals not only provides support and load-bearing functions but also enables various spatial movements, including flexion, extension, lateral bending, and torsion. The vertebrae on both sides of the torso joint also allow a limited range of relative motion. The lateral bending, axial rotation, and longitudinal pitching between the moving and fixed platforms of the bionic active torso mimic the lateral swaying, rolling, flexion, and extension of a canine’s torso during locomotion. This series of coordinated movements effectively replicates the motion patterns of mammalian torsos, with the potential to significantly enhance the flexibility and stability of quadrupedal robots.

### 2.2. Inverse Kinematics Analysis

In the entire parallel-configured torso structure, the motors on the static platform are arranged in pairs, evenly distributed at 120° intervals, as shown in Figure 3. Within each pair, the two motors are installed with a spacing of $D_{1}$ and an inclination angle of *η* with respect to the vertical plane. The output shaft of each motor is connected to a rocker at point $R^{S}$, forming a circular plane with a radius of $R_{1}$. The length of the rocker is *r*, and its distal end is connected to a spherical joint at point $Q^{S}$. The connecting rod, with a length of *L*, is attached to the upper spherical joint at point $Q^{M}$, which forms a circular plane with a radius of R. The spherical joints are grouped in pairs with a spacing of *D*, and the upper spherical joints are located at a height *h* from the moving platform. The distance from $R^{S}$ to the static platform is denoted as *h*. The coordinate systems of the moving and static platforms are defined as {*M*} and {*S*}, with their origins marked as $O^{M}$ and $O^{S}$, respectively. The parameters of the active torso mechanism are listed in Table 1 [21,23].

The vector $O^{S}$ to $R_{i}^{S}$ in the {*S*} coordinate system and the vector $O^{M}$ to $Q_{i}^{M}$ in the {*M*} coordinate system can be directly expressed as follows:(1)$$O^{S}R_{i}^{S}=[R_{1}\cos\omega_{i},R_{1}\sin\omega_{i},h_{1}]$$(2)$$O^{M}Q_{i}^{M}=[R\cos\sigma_{i},R\sin\sigma_{i},-h]$$

The angle $\omega_{i}$ represents the angle between $O^{S}R_{i}^{S}$ and the $X_{S}$ axis, while $\sigma_{i}$ represents the angle between $O^{M}Q_{i}^{M}$ and the $X_{M}$ axis. The parameters *α* and *β* are defined as follows: $a=arcsin\left(D_{1}/2R_{1}\right)$, $\beta =arccos\left(D/2R\right)$. If the desired pose of the moving platform relative to {*S*} is given as $\left[X,Y,Z,\phi ,\theta ,\psi \right]$ (representing position and orientation in roll, pitch, and yaw), the rotation matrix from coordinate frame {*M*} to {*S*} can be expressed as follows:(3)TMS=cφcθcψ−sφsψ−cφcθsψ−sφcψcφsθsφcθcψ+cφsψ−sφcθsψ+cφcψsθsφ−sθcψsθsψcθ

The position vector of $O^{S}$ relative to $Q_{i}^{M}$ in {*S*} can be written as follows:(4)OSQiM=TMS·OM·QiM+[X,Y,Z]T,(i=1,2…6)

The direction vector of the motor output shaft in the base coordinate frame {*S*} is given by the following equation:(5)$$\left\{\begin{array}{c} U_{i}=[\frac{h}{\tan\eta }\cos\zeta ,\frac{h}{\tan\eta }\sin\zeta ,h_{1}],(i=1,2,3\ldots 6)\\ \zeta =\left\{0,-\frac{2\pi }{3},-\frac{2\pi }{3},-\frac{4\pi }{3},-\frac{4\pi }{3},0\right\} \end{array}\right.$$

A new reference coordinate frame is established at the center of the rotational joint on the motor output side. The direction vectors of each axis in this frame are defined as follows:(6)$$X_{i}^{S}=\left\{\begin{array}{c} \frac{U_{i}\times Z^{S}}{\left|U_{i}||Z^{S}\right|},\;if\;i=1,3,5\\ \frac{Z^{S}\times U_{i}}{\left|U_{i}||Z^{S}\right|}\;,\;if\;i=2,4,6 \end{array}\right.,\;Y_{i}^{S}=\frac{U_{i}\times X_{i}^{S}}{\left|U_{i}||X_{i}^{S}\right|},\;Z^{S}=[0,0,1]$$

The vectors along each kinematic branch can then be expressed as follows:(7)$$\left\{\begin{array}{c} R_{i}^{S}Q_{i}^{S}=r(X_{i}^{S}\cos\delta_{i}+Y_{i}^{S}\sin\delta_{i})\\ R_{i}^{S}Q_{i}^{M}=O^{S}Q_{i}^{M}-O^{S}R_{i}^{S}\\ Q_{i}^{S}Q_{i}^{M}=R_{i}^{S}Q_{i}^{M}-R_{i}^{S}Q_{i}^{S} \end{array}\right.$$

By substituting into the above equation, we obtain the following equation:(8)$${\left|Q_{i}^{S}Q_{i}^{M}\right|}^{2}=L^{2}=r^{2}-2R_{i}^{S}Q_{i}^{M}\cdot R_{i}^{S}Q_{i}^{S}+{\left|R_{i}^{S}Q_{i}^{M}\right|}^{2}$$

To simplify the solution process, we define $\lambda_{i}=L^{2}-r^{2}-{\left|R_{i}^{S}Q_{i}^{M}\right|}^{2}$, based on (7) and (8), the following result can be obtained:(9)$$\lambda_{i}+2R_{i}^{S}Q_{i}^{M}\cdot r\left(X_{i}^{S}cos\delta_{i}+Y_{i}^{S}sin\delta_{i}\right)=0$$

Using the universal trigonometric identities:(10)$$\left\{\begin{array}{c} sin\delta_{i}=\left(2tan\left(\delta_{i}/2\right)\right)/\left(1+tan\left({\delta_{i}}^{2}/2\right)\right)\\ cos\delta_{i}=\left(1-tan\left({\delta_{i}}^{2}/2\right)\right)/\left(1+tan\left({\delta_{i}}^{2}/2\right)\right) \end{array}\right.$$

And defining $\mu =\tan\left(\delta_{i}/2\right)$, the equation can be further solved as follows:(11)$$\mu^{2}(\lambda_{i}-2rX_{i}^{S}\cdot R_{i}^{S}Q_{i}^{M})+\mu (4rY_{i}^{S}\cdot R_{i}^{S}Q_{i}^{M})+2rX_{i}^{S}\cdot R_{i}^{S}Q_{i}^{M}+\lambda_{i}=0$$

The motor rotation angle can thus be obtained as follows: $\delta_{i}=2arctan\left(\mu \right)$. This equation yields two solutions, from which the appropriate one can be selected based on the actual rotational requirements of the active torso. The entire solution process can therefore be simplified as follows:(12)$$\delta_{i}=f[X,Y,Z,\varphi ,\theta ,\psi ],i=1,2,\ldots 6$$

## 3. Bionic Active Torso Quadruped Robot

### 3.1. Bionic Quadruped Robot Structure

Inspired by the anatomical structure of canine quadrupeds, the bionic quadruped robot with an active torso developed in this study is illustrated in Figure 4. The robot is composed of the following four main components: the legs, front body, active torso, and rear body. The front and rear bodies fulfill three primary functions. First, they provide structural support by serving as attachment points for the legs. Second, the front body contains a spacious internal compartment for accommodating key hardware components, including the controller and battery. Third, they form the interface between the static and moving platforms of the torso mechanism, enabling a transition from a rigid torso structure to a biologically inspired active torso.

Notably, in the parallel-configured torso structure, the static platform—which houses the drive motors—is heavier than the moving platform. To maintain structural balance and enhance bionic fidelity, the static platform is connected to the front body, mimicking the skeletal layout observed in canine species. This structural arrangement improves both the stability and biological resemblance of the quadruped robot equipped with an active torso.

### 3.2. Single-Leg Kinematics

The kinematic modeling of the quadruped robot is conducted using the D-H method, as illustrated in Figure 5. The established coordinate systems are categorized into two sections: the right foreleg, representing the kinematic modeling of a single leg, and the right hind leg, which incorporates the active torso and hind leg kinematics. The structural parameters of the robot are listed in Table 2.

In the single-leg kinematics analysis, coordinate systems {2}, {3}, and {4} are defined at the root joint, hip joint, and knee joint of the right leg, respectively. These coordinate systems move along with the corresponding leg segments. Coordinate system {0} is fixed to the robot body, with its *X*-axis aligned with the robot’s forward direction, the *Z*-axis pointing upward perpendicular to the body, and the Y-axis perpendicular to the *XZ* plane, pointing to the left side of the robot.

For convenience in deriving the transformation matrices, coordinate system {1} is initially set to coincide with {2}. After deriving the transformation matrix in {1}, coordinate transformations are applied to obtain the transformation matrix in {0}. Additionally, coordinate system {5} is defined at the foot end and moves along with the foot.

The robot leg can be simplified as a series of interconnected links. Once coordinate frames are assigned to each link, the relative transformation between two adjacent coordinate frames, {*i* − 1} and {*i*}, can be established through two rotations and two translations. This transformation is commonly referred to as the generalized link transformation matrix and is defined as follows:(13)Tii−1=cθi−sθi0ai−1sθicαi−1cθicαi−1−sαi−1−disαi−1sθisαi−1cθisαi−1cαi−1dicαi−10001

In this matrix, *s* and *c* denote the sin and cos functions, respectively. The Denavit–Hartenberg (DH) parameters for a single leg are listed in Table 3.

Based on (13) and Table 3, the sequential transformations from coordinate frame {1} to {5} can be obtained as follows:(14)T21=c1−s100s1c10000100001    T32=c2−s20000−1L1s2c2000001T43=c3−s30L2s3c30000100001    T54=100L3010000100001

The transformation from coordinate frame {0} to frame {1} involves a 90° rotation about the *Y*-axis followed by a 180° rotation about the *X*-axis. The corresponding transformation matrix T10 is given by the following equation:(15)T10=Rot(x,180°)Rot(y,90°)=00100−10010000001

In summary, the overall transformation matrix T50 for a single leg can be expressed as follows:(16)T50=s23c230px−s1c23s1s23c1pyc1c23−c1s23s1pz0001

The first three elements of the last column of this transformation matrix represent the position of the foot end in coordinate frame {0}. Accordingly, the position of the right foreleg in frame {0} is given by the following equation:(17)$$\left\{\begin{array}{c} x=L_{2}s_{2}+L_{3}s_{23}\\ y=-L_{1}c_{1}-L_{2}s_{1}c_{2}-L_{3}s_{1}c_{23}\\ z=-L_{1}s_{1}+L_{2}c_{1}c_{2}+L_{3}c_{1}c_{23} \end{array}\right.$$

Similarly, the position of the left foreleg in coordinate frame {0} is given by the following equation:(18)$$\left\{\begin{array}{c} x=L_{2}s_{2}+L_{3}s_{23}\\ y=L_{1}c_{1}-L_{2}s_{1}c_{2}-L_{3}s_{1}c_{23}\\ z=L_{1}s_{1}+L_{2}c_{1}c_{2}+L_{3}c_{1}c_{23} \end{array}\right.$$

### 3.3. Kinematics of the Torso and Hind Limbs

A coordinate frame {0} is established at the robot’s center of mass, and a total of eight coordinate frames are defined along the path from the center of mass to the right hind leg’s foot. During the transformation from frame {0} to frame {1}, the primary contribution comes from the bionic parallel torso, which actively performs torso motions. This motion causes the rear body of the robot to swing laterally, resulting in a yaw angle $\theta_{r}$ between the *X*-axes of the two frames around the *Z*-axis. Simultaneously, translational displacements occur along the *X* and *Y* directions. Therefore, the transformation matrix T10 can be expressed as follows:(19)T10=cr−sr0−Lmcrsrcr0−Lmsr00100001

The transformation from coordinate frame {1} to frame {2} involves only a positional offset along the *Y*-axis. Specifically, the offset is $L_{n}$ for the left leg and $-L_{n}$ for the right leg. The corresponding transformation matrix T21 is given by the following equation:(20)T21=1000010−Ln00100001

The transformation from coordinate frame {2} to frame {3} involves a 90° rotation about the *Y*-axis, followed by a 180° rotation about the *X*-axis. The corresponding transformation matrix T32 is given by the following equation:(21)T32=Rot(x,180°)Rot(y,90°)=00100−10010000001

The transformation matrix from coordinate frame {3} to frame {7} is analyzed using the D-H method. The D-H parameters for a single leg are listed in Table 3.

The transformation from coordinate frame {3} to frame {7} can be derived by referring to the transformation from frame {1} to frame {5} in the single-leg kinematic model. The overall transformation matrix T73 is given by the following equation:(22)T73=c1c23−c1s23s1−L1s1+L2c1c2+L3c1c23s1c23−s1s23−c1L1c1+L2s1c2+L3s1c23s23c230L2s2+L3s230001

Here, $s_{i}$ and $c_{i}$ represent $sin(\theta_{i})$ and $cos(\theta_{i})$, respectively, while $s_{23}$ and $c_{23}$ denote $sin(\theta_{2}+\theta_{3})$ and $cos(\theta_{2}+\theta_{3})$. Accordingly, the final transformation matrix T70 in the torso *CoM* coordinate frame is given by the following equation:(23)T70=T10·T21·T32·T73T70=crs23+srs1c23crc23−srs1s23−src1crpx−srpy+srLn−crLmsrs23−crs1c23src23+crs1s23crc1srpx+crpy−crLn−srLmc1c23−c1s23s1pz0001

The first three elements of the last column of the transformation matrix represent the position coordinates of the foot end in coordinate frame {0}. Based on (17), (18) and (23), the position of the right hind leg in frame {0} can be obtained as follows:(24)$$\left\{\begin{array}{c} x=c_{r}(L_{2}s_{2}+L_{3}s_{23})-s_{r}(-L_{1}c_{1}-L_{2}s_{1}c_{2}-L_{3}s_{1}c_{23})+s_{r}L_{n}-c_{r}L_{m}\\ y=s_{r}(L_{2}s_{2}+L_{3}s_{23})+c_{r}(-L_{1}c_{1}-L_{2}s_{1}c_{2}-L_{3}s_{1}c_{23})-c_{r}L_{n}-s_{r}L_{m}\\ z=-L_{1}s_{1}+L_{2}c_{1}c_{2}+L_{3}c_{1}c_{23} \end{array}\right.$$

The position of the left hind leg in coordinate frame {0} can be obtained as follows:(25)$$\left\{\begin{array}{c} x=c_{r}(L_{2}s_{2}+L_{3}s_{23})-s_{r}(L_{1}c_{1}-L_{2}s_{1}c_{2}-L_{3}s_{1}c_{23})-s_{r}L_{n}-c_{r}L_{m}\\ y=s_{r}(L_{2}s_{2}+L_{3}s_{23})+c_{r}(L_{1}c_{1}-L_{2}s_{1}c_{2}-L_{3}s_{1}c_{23})+c_{r}L_{n}-s_{r}L_{m}\\ z=L_{1}s_{1}+L_{2}c_{1}c_{2}+L_{3}c_{1}c_{23} \end{array}\right.$$

### 3.4. Model Predictive Controller Based on Shifted Center of Mass Position

Model predictive control (*MPC*) is essentially a rolling optimization problem over a finite time horizon and has been increasingly applied in motion control algorithms for quadruped robots in recent years. Due to the repetitive switching of the robot’s legs between stance and swing phases, the foot-end of the robot is susceptible to irregular impacts caused by various factors, making it difficult to maintain balance. Furthermore, the rapid changes in contact states among the legs add additional complexity to the control problem. To address these challenges, this paper employs a linearized model predictive controller for quadruped robots. By leveraging discrete finite-horizon predictive control, the controller calculates the desired contact forces at the feet, which are then mapped to joint torques. As a result, the desired joint control torques are obtained.

#### 3.4.1. Simplified Dynamic Model

At present, most conventional quadruped robots adopt model predictive control (*MPC*) based on a single rigid body model [24]. However, when it comes to quadruped robots with a spine structure, the motion of the spine introduces additional complexity to the system, leading to significant variations in the robot’s center of mass and inertial parameters. Traditional approaches typically simplify the entire robot as a single rigid body with constant inertial properties, neglecting the changes induced by spinal motion. Applying such methods directly to the bionic parallel torso quadruped robot proposed in this study may result in unstable motion and even cause the robot to fall. Therefore, to ensure stable and efficient locomotion in complex environments, this paper proposes a rigid body model that accounts for the influence of spinal motion. Specifically, the bionic parallel torso quadruped robot is simplified as a single rigid body dynamic model with variable inertial parameters.

The quadruped robot is modeled as a single rigid body with a variable inertia tensor. Considering the design of the robot, in which the legs are relatively lightweight and have low inertia, the leg masses are neglected in the model. As a result, the system is simplified to a rigid body with variable inertial parameters and four massless legs, forming a single rigid body torso model of the quadruped robot.

As shown in Figure 6, the world frame is defined as {*W*}, the inertial frame is defined as {*I*} and the body frame is defined as {*B*}. The vector $r_{i}$ denotes the position of the *i*-th foot relative to the center of mass of the body, and $f_{i}$ represents the ground reaction force acting on the *i*-th foot.

The quadruped robot is subjected to gravitational force and ground reaction forces. Accordingly, the rigid body dynamics of the robot in the world frame can be expressed as follows:(26)p¨cW=∑i=14fiWm−g

In the above equation, pcW represents the position of the robot’s center of mass in the frame {*W*}, *m* denotes the total mass of the quadruped robot, and *g* is the gravitational acceleration vector. fiW is the contact force applied at the foot-end of the *i*-th leg in the frame {*W*}.

According to the law of conservation of angular momentum, the rotational dynamics of the robot’s body can be expressed as follows:(27)∑i=14riI×fiW+03×1×(m·g)=ddt(IWωW)ddt(IWωW)=IW·ω˙W+ωW·I˙W

In this equation, ωW denotes the angular velocity of the robot’s body in the frame {*W*}, IW represents the inertia tensor of the robot described in the frame {*W*}, and riI is the position vector of the support point of the *i*-th leg relative to the body frame {*I*}.

Given that the robot’s angular velocity is relatively small and has minimal impact on the system dynamics, the nonlinear terms can be removed to simplify the model. Based on (27), the following expression is derived:(28)ddt(Iwww)≈Iw×w˙w=∑i=14riI×fiw

The relationship between the inertia tensors expressed in the body-fixed frame {*B*} and world frame {*W*} can be described as follows:(29)Iw=RBw·IB·RBTw

Due to the presence of an animal-like torso structure in the bionic parallel torso quadruped robot, the inertial parameters of the robot change with torso motion. In this study, the front and rear body segments of the robot are simplified as rigid bodies with lumped masses, connected via a parallel-linked torso structure. To further simplify the model, each segment of the torso is represented as a point mass connected by massless rods. The simplified model for computing the torso’s overall inertia is illustrated in Figure 7. The overall torso inertia of the robot, denoted as IW, can be expressed as follows:(30)Iw=I+IF+IH+mFdF+mHdH

Here, $m_{F}$ and $m_{H}$ represent the masses of the front and rear bodies of the robot. $l_{F}$ and $l_{H}$ denote the distances from the front and rear bodies to the moving and static platforms of the parallel torso mechanism; $\theta_{F}$ and $\theta_{H}$ are the rotation angles of the front and rear bodies with respect to the parallel torso. $d_{F}$ and $d_{H}$ represent the relative distances between the front/rear bodies and the center of mass of the parallel torso mechanism and are expressed as follows:(31)$$\left\{\begin{array}{c} d_{F}=\sqrt{{l_{1}}^{2}+{\left(l_{F}\right)}^{2}-2\cdot l_{1}l_{F}cos(\theta_{F})}\\ d_{H}=\sqrt{{l_{2}}^{2}+{\left(l_{H}\right)}^{2}-2\cdot l_{2}l_{H}cos(\theta_{H})} \end{array}\right.$$

In the single rigid body model, the robot’s orientation is represented using *Z-Y-X* Euler angles $\Theta =\left[\varphi ,\theta ,\psi \right]$, where φ is the roll angle, θ is the pitch angle, and ψ is the yaw angle. Assuming that the body-fixed coordinate frame initially aligns with the world frame in both position and orientation. The corresponding rotation matrix can be expressed as follows:(32)$$R=R_{z}(\psi )\cdot R_{y}(\theta )\cdot R_{x}(\varphi )$$

Therefore, the angular velocity in the world coordinate frame is expressed using the first-order time derivatives of the *Z-Y-X* Euler angles. The expression is given by [25] the following equation:(33)ωw=RΘ˙=cos(θ)cos(φ)−sin(φ)0cos(θ)sin(φ)cos(φ)0−sin(θ)01Θ˙

By combining the translational and rotational dynamics of the quadruped robot’s single rigid body, a complete first-order state-space equation can be formulated, as shown below. In this expression, $0_{3}$ denotes a 3 × 3 zero matrix, and $I_{3}$ denotes a 3 × 3 identity matrix. The resulting dynamic model of the quadruped robot is established as follows:(34)ddtΘpWωp˙W=0303R−103030303130303030303030303ΘpWωp˙W+0303030303030303I−1W·r1×II−1W·r2×II−1W·r3×II−1W·r4×I13/m13/m13/m13/mf1Wf2Wf3Wf4W+030303g

To formulate a standard first-order state-space representation, the gravitational acceleration *g* is introduced as an additional state variable. Accordingly, (34) can be reformulated as follows:(35)ddtΘpWωp˙Wg︸X˙(t)=0303R−103003030313003030303003030303101×301×301×301×30︸ACΘpWωp˙Wg︸X(t)+0303030303030303I−1W·r1×II−1W·r2×II−1W·r3×II−1W·r4×I13/m13/m13/m13/m01×301×301×301×3︸BCf1Wf2Wf3Wf4W︸U(t)

The control system of the quadruped robot operates in a discrete-time domain. Therefore, the discretized state-space equation can be expressed as follows:(36)$$\frac{X(h+1)-X(h)}{dt}=A_{c}X(h)+B_{c}U(h)$$

Here, $X_{h}$ and $X_{h+1}$ represent the robot’s pose state at time steps h and h+1, respectively. Accordingly, the discrete-time state-space equation of the quadruped robot can be formulated in the following form:(37)$$X_{h+1}=A_{h}X_{h}+B_{h}U_{h}$$

Given the known control input $U_{h}$ at the *h*-th control cycle, the subsequent state $X_{h+1}$ can be propagated from $X_{h}$. By recursively applying (37), the future states over *k* cycles can be predicted. The model predictive state equation can be formulated as follows:(38)$$\underset{X_{ref}}{\underset{︸}{\left[\begin{array}{c} X_{1}\\ X_{2}\\ \vdots \\ X_{k} \end{array}\right]}}=\underset{A_{qp}}{\underset{︸}{\left[\begin{array}{c} A_{0}\\ A_{1}A_{0}\\ \vdots \\ \prod_{i=k-1}^{0}A_{i} \end{array}\right]}}X_{0}+\underset{B_{qp}}{\underset{︸}{\left[\begin{array}{cccc} B_{0} & 0 & \ldots & 0\\ A_{1}B_{0} & B_{1} & \ldots & 0\\ \vdots & \vdots & \ddots & 0\\ (\prod_{j=k-1}^{1}A_{j})B_{0} & (\prod_{j=k-1}^{2}A_{j})B_{1} & \ldots & B_{k-1} \end{array}\right]}}\underset{U_{eq}}{\underset{︸}{\left[\begin{array}{c} U_{0}\\ U_{1}\\ \vdots \\ U_{k-1} \end{array}\right]}}b$$

Here, $X_{ref}$ denotes the state matrix over the future *k* control cycles. $A_{eq}$ is the state transition matrix, $B_{eq}$ is the control input matrix, and $U_{eq}$ represents the control input matrix within the prediction horizon *k*.

#### 3.4.2. Construction of the Model Predictive Controller (MPC)

In this section, *MPC* is developed for the quadruped robot based on the shifted center of mass position. The objective of the *MPC* is to predict the optimal desired contact forces at the feet that ensure stable locomotion of the robot. Given the current discrete-time state-space model of the system, the controller estimates the future states over a finite prediction horizon. At each sampling instant, an optimal control problem with constraints is solved over this finite horizon. The resulting optimal control sequence is then obtained, and only the first control input of this sequence is applied as the output of the controller to govern the desired input for the supporting legs in the next control cycle. This process is repeated at every control step, enabling rolling (receding horizon) optimization of the control inputs.

In this study, the problem is formulated as an *MPC* problem that solves for the optimal control inputs over a prediction horizon of N steps. Let $X_{h+1}$ denote the system state at time step h+1, and the corresponding reference state $X_{h+1}^{\mathrm{ref}}$. The objective is to minimize the tracking error between the system and the reference states. This goal is achieved by solving the following optimization problem:(39)$$\begin{array}{c} \underset{U}{\min}\sum_{h=0}^{N-1}\left\|X_{h+1}-X_{h+1}^{ref}\right\|Q_{h}+\left\|U_{h}^{P}\right\|R_{h}\\ s.t.\;X_{h+1}=A_{h}X_{h}+B_{h}U_{h}\\ \;q_{\min}\leq U_{h}\leq q_{\max}\\ \;FU_{h}\leq 0,F=\left[\begin{array}{ccc} 1 & 0 & -\mu \\ -1 & 0 & -\mu \\ 0 & 1 & -\mu \\ 0 & -1 & -\mu \end{array}\right] \end{array}$$

Here, $U_{h}^{P}=[U_{1},U_{2},\ldots ,U_{h}]$ denotes the control input matrix over the prediction horizon. $Q_{h}$ is the state weighting matrix that ensures adequate tracking performance, while $R_{h}$ is the control weighting matrix that penalizes excessive control effort. F represents a linearized approximation of the Coulomb friction cone, enforcing that the foot-end contact forces remain within the friction limits, where μ is the friction coefficient between the foot and the ground. $q_{\min}$ and $q_{\max}$ define the lower and upper bounds of the control input, respectively. These bounds help reduce energy consumption and impulsive forces, thereby enhancing control robustness and preventing issues such as actuator saturation or motor overheating.

Based on (38) and (39), the optimization problem can be formulated in the standard form of a quadratic programming (*QP*) problem:(40)$$\underset{U}{\min}\frac{1}{2}U^{T}\underset{H}{\underset{︸}{\left\{2(B_{\mathrm{eq}}^{T}\cdot Q\cdot B_{eq}+R)\right\}}}U+U^{T}\underset{g}{\underset{︸}{\left\{2\cdot B_{\mathrm{eq}}^{T}Q(A_{eq}X_{0}-X_{ref})\right\}}}$$

In this study, consistent weighting parameters are employed across all control cycles.(41)$$\left\{\begin{array}{c} Q.diag(13\times h,13\times h)=Q_{h}\\ R.diag(12\times h,12\times h)=R_{h} \end{array}\right.$$

By solving the above linear optimization problem, a set of foot-end virtual forces that satisfy the corresponding constraints can be obtained as the optimal control input for the current control cycle.(42)$$U_{i}=f^{MPC}$$

Here, $f^{MPC}$ denotes the foot-end reaction force during the support phase, computed based on the *MPC* method. Accordingly, the joint torques τ of each leg generated by the *MPC* controller can be expressed as follows:(43)$$\tau =R^{T}\cdot J^{T}\cdot f^{MPC}$$

Here, R denotes the rotation matrix of the robot, and J represents the Jacobian matrix.

#### 3.4.3. Overall Motion Control

The previous subsection established a rigid body dynamic model of the quadruped robot with variable inertial parameters and designed *MPC* to predict the optimal contact forces at the feet. These forces are then translated into motor torque commands through coordinated leg and torso motion control based on joint variables.

The overall control framework of the bionic parallel torso quadruped robot is illustrated in Figure 8. The control input is defined as the desired velocity of the robot. Based on this command, a motion planning module generates a desired trajectory for the robot’s body. Motor joint angles and angular velocities are used as inputs to compute the necessary state variables for control using both leg and torso kinematics. Together with the robot’s structural parameters, the relative positions of various components are calculated to obtain the position and orientation of the torso and body.

The model predictive controller then utilizes the simplified dynamic model to optimize the control inputs and predict the optimal contact forces at the feet over the prediction horizon. These predicted forces, combined with leg-level *PD* control, yield the required joint torques, which are subsequently applied to the robot’s legs. Motor feedback is used to close the control loop, forming a simple yet effective feedback control system.

## 4. Torso–Leg Coordinated Locomotion

### 4.1. Torso–Leg Coordinated Locomotion Mechanism

The diagonal trotting gait of the quadruped robot with a bionic active torso is designed based on behavioral observations of four-legged animals. This gait essentially mimics the natural movement patterns of quadrupedal animals. To effectively plan the robot’s trotting behavior, it is essential to analyze the key motion features exhibited by typical quadrupeds, such as canines, during diagonal trotting. As illustrated in Figure 9, the analysis focuses on the coordinated movement of the dog’s torso during this gait.

It can be observed that the motion of the legs always occurs in diagonally paired limbs, accompanied by a torso motion that drives both the shoulder and hip regions. This motion is synchronized with the forward swing of the legs. Therefore, the design of the diagonal trotting gait in this study fully incorporates the yaw motion of the torso and its coordination with the legs, ensuring stable and balanced locomotion of the robot.

The coordination and phase relationship between the torso and hind legs are shown in Figure 10. In Figure 10a, a novel trotting gait is presented by superimposing active yaw motion of the torso on a conventional diagonal trot. The robot’s forebody moves forward according to the planned trajectory, and the front legs follow the designated foot-end paths. Meanwhile, the rear body undergoes lateral oscillation induced by the active yaw motion of the torso. This oscillation causes the rear legs to deviate from the motion pattern of the front legs. Therefore, adjustments to the joint angles of the rear legs are required to ensure that their motion conforms to the desired swing and support trajectories while also aligning the touchdown points of the front and rear legs on the same side along a straight line.

As shown by the red line in the figure, the torso motion causes a shift in the robot’s center of mass (*CoM*). When the torso yaws, the longitudinal axis of the torso deviates from a straight line to a curved path, shifting the *CoM* laterally from the center toward one side of the body. When the torso yaws in the opposite direction, the *CoM* shifts to the other side.

Figure 10b illustrates the phase transition pattern that governs the coordinated motion between the active torso and the rear legs. The torso motion is divided into four stages: forward yaw, forward limit, reverse yaw, and reverse limit. Throughout this process, phase transition rules are applied to synchronize the motion of the torso and legs. Additionally, phase transitions are triggered based on the torso’s motion limits and the ground contact states of the rear legs (touchdown and liftoff).

### 4.2. CPG-Based Torso Motion Trajectory

In recent years, inspired by neurobiological principles, the Central Pattern Generator (*CPG*) has been widely adopted in robotic control. The fundamental function of the *CPG* is to produce rhythmic behaviors such as locomotion, respiration, sucking, and chewing [26]. These functions have been extensively studied by biologists and neuroscientists. For example, multi-degree-of-freedom robots can achieve motion control through *CPG*s, exhibiting high levels of coordination and adaptability [27,28].

In this study, a highly coupled Hopf oscillator network is employed to generate periodic motion signals for the active torso of a quadruped robot [21,29,30]. In the leg *CPG* system, each leg is driven by an individual oscillator. To realize coordinated motion between the legs and the torso, the torso is modeled as a virtual fifth leg. By adjusting the phase parameters within the *CPG* network, a dedicated *CPG* signal for the torso is generated. The network structure of the proposed *CPG* system is illustrated in Figure 11.

Based on the inter-leg phase relationships in angular (diagonal) gait and the phase coordination required for synergistic torso–leg motion, the phase differences among the four legs, as well as between the torso and each leg, are defined as follows:(44)$$\left\{\begin{array}{c} \left[\begin{array}{cccc} \Delta_{12} & \Delta_{23} & \Delta_{34} & \Delta_{41} \end{array}]=[\begin{array}{cccc} -\pi & \pi & -\pi & \pi \end{array}\right]\\ \left[\begin{array}{cccc} \Delta_{15} & \Delta_{25} & \Delta_{35} & \Delta_{45} \end{array}]=[\begin{array}{cccc} -\frac{3}{5}\pi & \frac{2}{5}\pi & -\frac{3}{5}\pi & \frac{2}{5}\pi \end{array}\right] \end{array}\right.$$

Here, $\Delta_{ij}$ denotes the phase difference between Leg *i* and Leg *j* (*i*,*j* = 1, 2, 3, 4, 5), where Legs 1 to 4 correspond to the right foreleg, right hind leg, left hind leg, and left foreleg, respectively.

### 4.3. Torso Trajectory Based on the Center of Mass

When the robot is walking or running, *CoM* should remain within a stable support polygon to prevent tilting or falling. To facilitate *CoM* calculation, the robot’s total mass is assumed to be concentrated at the geometric centers of its major components. These centers are denoted as $m_{i}\left(i=0,1,2,3,4\right)$, as shown in Figure 12.

A reference coordinate system is established at $m_{0}$, with the X-axis aligned with the robot’s forward direction and the Y-axis perpendicular to the body, pointing to the left. The distance from point $m_{0}$ to $m_{1}$ is denoted as $L_{m1}$, from $m_{1}$ to $m_{2}$ as $L_{m2}$, and so on. In the computation, distances are assigned negative values according to their direction along the X-axis. The center of mass is calculated as follows:(45)$$\left\{\begin{array}{c} x_{COG}=\frac{m_{0}\times 0+m_{1}Lm_{1}+m_{2}(\sum_{i=1}^{2}Lm_{i})+m_{3}(\sum_{i=1}^{2}Lm_{i}+Lm_{3}cosq_{r})+m_{4}(\sum_{i=1}^{2}Lm_{i}+cosq_{r}\sum_{i=3}^{4}Lm_{i})}{m_{0}+m_{1}+m_{2}+m_{3}+m_{4}}\\ y_{COG}=\frac{(m_{0}+m_{1}+m_{2})\times 0+m_{3}Lm_{3}sinq_{r}+m_{4}(Lm_{3}+Lm_{4})sinq_{r}}{m_{0}+m_{1}+m_{2}+m_{3}+m_{4}} \end{array}\right.$$

Here, $\theta_{r}$ denotes the yaw angle of the torso. Since the rotation of the active torso does not affect the robot’s position along the Z-axis, the coordinates of the center of mass in the body coordinate frame can be expressed as follows:(46)$$COG=[x_{COG},y_{COG},0]$$

To generate the torso motion trajectory, the yaw angle of the torso is determined by maintaining the robot’s center of mass along the support axis [31]. When the left foreleg and right hind leg are in the support phase, the positions of their respective foot ends are given by $\left(L_{r},L_{k}\right)$ and $\left(-2L_{m}+L_{r},-L_{k}\right)$, where $L_{m}$ denotes the distance from the torso center of mass to the front and rear hip joints. The support axis can thus be expressed as follows:(47)$$y=\frac{L_{k}}{L_{m}}x+\frac{L_{k}}{L_{m}}(L_{m}-L_{r})$$

As the active torso begins to swing toward the left side of the robot’s body, the *CoM* can be simplified from (45) as follows:(48)$$\left\{\begin{array}{c} x_{COG}=\frac{X_{L}+X_{S}\cos\theta }{m_{z}}\\ y_{COG}=\frac{Y_{L}+Y_{S}\sin\theta }{m_{z}}\\ X_{L}=-Ls_{0}m_{0}-Ls_{1}m_{1}-(Ls_{1}+Ls_{2})(m_{2}+m_{3}+m_{4})\\ X_{S}=-Ls_{3}m_{3}-(Ls_{3}+Ls_{4})m_{4}\\ \begin{array}{c} Y_{L}=Ls_{0}(m_{0}+m_{1}+m_{2})\\ Y_{S}=Ls_{3}m_{3}+(Ls_{3}+Ls_{4})m_{4}\\ m_{z}=m_{0}+m_{1}+m_{2}+m_{3}+m_{4} \end{array} \end{array}\right.$$

The yaw angle of the active torso is given by the following equation:(49)$$\theta =2\varsigma \arctan\frac{L_{m}Y_{s}-\sqrt{L_{m}^{2}Y_{s}^{2}-{\left[L_{k}X_{L}-Y_{L}L_{m}+L_{k}m_{z}(L_{m}-L_{r})\right]}^{2}+L_{k}^{2}X_{s}^{2}}}{L_{k}(X_{L}-X_{s})-Y_{L}L_{m}+L_{k}m_{z}(L_{m}-L_{r})}$$

Here, ς represents a tuning coefficient used to regulate the influence of inertial forces induced by torso yaw motion in the quadruped robot with an active torso. When the right foreleg and left hind leg enter the support phase, the positions of their foot ends are given by $\left(L_{r},-L_{k}\right)$ and $\left(-2L_{m}+L_{r},L_{k}\right)$, respectively. The corresponding support axis can then be expressed as follows:(50)$$y=-\frac{L_{k}}{L_{m}}x+\frac{L_{k}}{L_{m}}(L_{r}-L_{m})$$

The yaw angle of the torso is given by the following equation:(51)$$\theta =2\varsigma \arctan\frac{-L_{m}Y_{s}+\sqrt{L_{m}^{2}Y_{s}^{2}-{\left[-L_{k}X_{L}-Y_{L}L_{m}+L_{k}m_{z}(L_{r}-L_{m})\right]}^{2}+L_{k}^{2}X_{s}^{2}}}{L_{k}(X_{L}-X_{s})-Y_{L}L_{m}+L_{k}m_{z}(L_{m}-L_{r})}$$

### 4.4. Torso Motion Trajectory Based on Zero Moment Point

Compared to static stability methods, the Zero Moment Point (*ZMP*) approach [32] enables the robot to maintain dynamic balance while achieving a certain level of locomotion speed. The coordinate expression of the *ZMP* is given as follows:(52)$$\left\{\begin{array}{c} x_{ZMP}=x_{COG}-\frac{z_{COG}{\ddot{x}}_{COG}}{g}\\ y_{ZMP}=y_{COG}-\frac{z_{COG}{\ddot{y}}_{COG}}{g} \end{array}\right.$$

Here, $x_{COG}$, $y_{COG}$, and $z_{COG}$ denote the coordinates of the robot’s *CoM* in the *X*, *Y*, and *Z* directions, respectively. ${\ddot{x}}_{COG}$ and ${\ddot{y}}_{COG}$ represent the accelerations of the *CoM* along the *X* and *Y* axes, and g is the gravitational acceleration. The position of the *ZMP* can be determined based on the *CoM* coordinates, gravitational acceleration, and ground height.

In contrast to the *CoM*-based trajectory planning approach to ensure dynamic stability during the planned diagonal trotting gait of the quadruped robot with an active torso, the *ZMP* must remain on the support line at all times. By differentiating the (45), the accelerations of the center of mass can be derived and subsequently substituted into (52), yielding:(53)$$\left\{\begin{array}{c} x_{ZMP}=\frac{X_{L}}{m_{z}}+(1+\frac{H}{g})\frac{X_{S}}{m_{z}}\cos\theta \\ y_{ZMP}=\frac{Y_{L}}{m_{z}}+(1+\frac{H}{g})\frac{Y_{S}}{m_{z}}\sin\theta \end{array}\right.$$

When the left foreleg and right hind leg enter the support phase, the support axis is defined according to (47). By substituting the position of the *ZMP* into the equation, the following result can be obtained:(54)$$\left\{\begin{array}{c} \theta =2\varsigma \arctan\frac{L_{Y}-\sqrt{L_{Y}^{2}-L_{L}^{2}+L_{H}^{2}}}{L_{L}-L_{H}}\\ L_{L}=L_{k}X_{L}g+L_{k}m_{z}g(L_{m}-L_{r})-Y_{L}gL_{m}\\ L_{Y}=L_{m}(g+h)Y_{S}\\ L_{H}=L_{k}X_{S}(g+h) \end{array}\right.$$

When the right foreleg and left hind leg enter the support phase, the support axis is defined according to (50). By substituting the position of the Zero Moment Point (*ZMP*) into the equation, the following result is obtained:(55)$$\theta =2\varsigma \arctan\frac{-L_{Y}+\sqrt{L_{Y}^{2}-L_{L}^{2}+L_{H}^{2}}}{L_{L}-L_{H}}$$

### 4.5. Algorithm Verification Framework for Bionic Active Torso Quadruped Robot

To validate the effectiveness of the proposed motion control and trajectory planning algorithms for the bionic quadruped robot with an active torso, a co-simulation framework based on MATLAB 2021b and MSC.ADAMS 2019 was established, as illustrated in Figure 13. The control system is composed of four main modules: gait control, leg motion control, torso trajectory generation, and torso motion control.

In the gait control module, gait parameters such as the gait period and duty ratio are first defined. Then, *CPG* signals are generated using Hopf oscillators as a function of time *t*, which are used to plan the motion trajectories of the robot’s foot ends. The leg motion control module incorporates both impedance control and gravity compensation. The desired foot-end positions are obtained from the planned trajectories, while the actual positions are computed through forward kinematics using joint angle feedback. The deviation between the desired position *P*_*t*_ and the actual position *P*_*a*_ is processed by an impedance controller to compute a virtual contact force *F*. This force, combined with gravity compensation, is then mapped to joint torques via the leg Jacobian matrix and applied to the virtual prototype of the quadruped robot with an active torso. The torso trajectory generation module supports three different strategies: trajectories based on *CPG* signals, *CoM* trajectories, and trajectories derived from the *ZMP* criterion. These methods allow for flexible and adaptive torso motion planning under various locomotion scenarios. In the torso motion control module, the desired trajectories are converted into motor rotation angles via inverse kinematics, which are then applied to the active torso joints. Closed-loop *PID* control is implemented based on real-time pose feedback from the ADAMS 2019 virtual prototype to minimize the position and orientation errors of the moving platform, thereby achieving more accurate and stable torso motion.

This co-simulation framework enables high-fidelity validation of the proposed control algorithms under realistic mechanical and environmental conditions, providing a reliable foundation for further development of bionic quadruped locomotion systems.

## 5. Simulation and Analysis

### 5.1. Motion Performance of the Bionic Parallel Torso Quadruped Robot

To evaluate the influence of bionic parallel torso motion on overall locomotion performance, a set of comparative simulations was conducted under the trot gait. The experiments compare the robot’s motion and pose parameters in two conditions: one with a fixed bionic parallel torso and the other with active torso motion (pitching movement). These simulations provide an intuitive understanding of how active torso motion affects the dynamic behavior and stability of the quadruped robot.

During the flat-ground trot gait simulation, the robot was assigned a forward velocity of *v* = 0.3 m/s, and the total simulation duration was set to 10 s. In both test conditions—fixed torso and active torso—a consistent gait pattern and control strategy were applied to ensure fair comparison.

Figure 14 illustrates the simulation results of the quadruped robot performing a Trot gait with a fixed bionic parallel torso. From the first to the fourth subfigure, the robot’s diagonal gait pattern can be observed, characterized by alternating leg movements and periodic transitions. These snapshots demonstrate how the quadruped achieves stable locomotion on flat terrain even when the torso remains static.

The simulation results of the quadruped robot performing diagonal (trot) gait with active torso pitching motion are shown in Figure 15. From the first to the sixth subfigure, the robot’s locomotion process under coordinated leg-torso movement can be observed. In the first to third subfigures, the bionic parallel torso transitions from a horizontal posture to its maximum downward pitch angle and then returns to the horizontal state. In the fourth to sixth subfigures, the torso moves into an upward-pitched posture, reaching its maximum angle before returning to the neutral horizontal position. Throughout the trot gait, the robot achieves coordinated motion between the active torso and the leg movements, consistent with the intended control behavior. It can be observed that even with torso pitching motion, the quadruped robot maintains stable locomotion, demonstrating the effectiveness of the proposed motion control strategy in simulation.

As shown in Figure 16, by observing the displacement of the robot’s *CoM* in both simulation cases, the actual forward speed in the X-direction is calculated. When the bionic parallel torso is active, the robot’s actual speed is approximately 0.286 m/s, which differs by about 4.6% from the desired speed. In contrast, when the torso is fixed, the actual speed is about 0.275 m/s, resulting in a speed error of approximately 8.3% compared to the expected value. This comparison indicates that the error in the forward speed of the robot is smaller when the bionic parallel torso is in motion.

In the Y-direction, the displacement of the robot’s *CoM* is relatively small during torso motion, indicating a more stable locomotion state. In the Z-direction, the *CoM* oscillates within a certain range in both simulation cases, with no significant difference between the curves, reflecting a stable movement state.

As shown in Figure 17, by observing the variation of the robot’s pitch angle in two simulation scenarios, it is evident that the robot’s walking tends to stabilize after approximately 3 s. The pitch angle exhibits a regular and convergent pattern over time. Due to the introduction of bionic parallel torso motion in the pitch direction, the variation in pitch angle during walking is more pronounced compared to the case with a fixed torso.

As shown in Figure 18, the variation in the vertical (Z-direction) ground reaction forces at the foot ends is presented for both simulation scenarios. After the robot reaches a stable walking state, the amplitude of the force in the fixed torso case (blue curve) is approximately 290 N, while in the bionic parallel torso motion case (red curve), the amplitude is around 278 N. This comparison indicates that the incorporation of bionic parallel torso motion reduces the foot-end impact forces during walking, thereby contributing to improved walking stability and smoother ground contact.

### 5.2. Coordinated Motion of the Bionic Parallel Torso Quadruped Robot

The simulation of the bionic parallel torso quadruped robot focuses on two conditions under the trot gait: a rigid (fixed) torso and an actively active torso. In the rigid torso configuration, the robot is modeled with a fixed torso, and the motion cycle is set to 0.5 s. The simulated motion of the quadruped robot with a fixed torso is illustrated in Figure 19. To better visualize leg motion, the right foreleg and left hind leg are marked in black. As shown in the first frame, the right foreleg and left hind leg are in the swing phase, while the left foreleg and right hind leg are in the stance phase. In the second frame, the right foreleg and left hind leg continue to swing forward. In the third frame, they complete the swing and enter the stance phase, while the left foreleg and right hind leg begin their swing phase. In the final frame, the left foreleg and right hind leg continue swinging forward. Through periodic and alternating leg movements, the robot achieves diagonal (trot) gait locomotion under the fixed torso condition.

The simulated motion with an active bionic parallel torso during trot gait is shown in Figure 20. In the first frame, the front and rear body segments are aligned parallel along the horizontal axis, and the torso’s yaw angle is 0°. As the robot moves forward, the torso begins to yaw to the right, following the swing motion of the right foreleg and left hind leg. When these legs complete their swing and enter the stance phase, the torso yaws to the left, passing through the aligned configuration again (third frame), and continues to rotate until reaching the maximum yaw angle to the left, at which point the left foreleg and right hind leg complete their swing phase and start supporting. This coordinated torso-leg motion during diagonal gait demonstrates the effectiveness of the bionic parallel torso in achieving dynamic adaptability.

The workspace of the foot-end is computed based on the actual joint angles of the right hind leg during torso motion, as shown in Figure 21. The red region represents the workspace under the fixed torso condition, while the blue region corresponds to the torso 7.39° based on the center of mass (*CoM*) trajectory. The area of the blue region is approximately 54% larger than that of the red region. Since the foot trajectory primarily involves forward motion in the X-direction and lifting in the Z-direction, only the XZ-plane projection is presented. Moreover, the difference in torso yaw amplitudes under different trajectory strategies is small, so only the *CoM*-based workspace is shown here. The results demonstrate that the torso yaw motion significantly expands the foot-end workspace, thereby improving the flexibility and range of motion of the quadruped robot.

During trot gait locomotion, excessive lateral deviation may lead to loss of balance, especially on uneven terrain. In addition, the stepping pattern of the trot gait can induce body rolling, causing significant tilt of the robot during locomotion. Therefore, it is essential to minimize both lateral deviation and body inclination to maintain stable movement. As illustrated in Figure 22a, the Y-direction displacement of the robot body under different torso yaw trajectories is compared with that of a fixed-torso configuration. The four curves represent Rigid torso 1 (fixed torso), Torso track 2 (*CPG*-based trajectory), Torso track 3 (*CoM*–based trajectory), and Torso track 4 (*ZMP*-based trajectory).

In the first 2 s, before the torso motion is applied, all curves exhibit consistent trends. Between 2 and 6 s, the Y-direction displacement starts to diverge among the four configurations; however, the curves remain close and cross over each other without significant separation. At 6 s, the Y-direction displacements for curves 1–4 are −0.0316 m, −0.0320 m, −0.0268 m, and −0.0351 m, respectively. Compared to the fixed-torso case (curve 1), the lateral deviations are reduced by 1.27%, 15.19%, and −11.08% for curves 2, 3, and 4, respectively. From 6 to 10 s, curves 2 and 3 show smaller variations and clear separation from curves 1 and 4. Curves 1 and 4 exhibit similar trends and larger Y-direction displacements. This indicates that torso yawing motion can effectively reduce lateral deviation, particularly with center-of-mass-based trajectory planning. At 10 s, the Y-direction displacements are −0.0613 m (curve 1), −0.0495 m (curve 2), −0.0419 m (curve 3), and −0.0600 m (curve 4), representing reductions of 19.25%, 31.65%, and 2.12%, respectively, compared to curve 1. Overall, the center-of-mass-based torso trajectory achieves the best performance in reducing lateral deviation, while the *ZMP*-based trajectory exhibits minimal improvement over the fixed-torso case. The relatively poor performance of the *ZMP*-based method is likely due to the asymmetry in yaw motion, which causes an imbalance in left and right rotation amplitudes of the moving platform.

Figure 22b presents the pitch angle comparison under different torso trajectories, while Figure 22c provides a magnified view of the pitch angle variation between 3 and 4 s. In the first 2 s, before torso motion is introduced, all four curves follow an identical trend. After 2 s, although the curves maintain similar shapes, their amplitudes differ. At t = 3.1 s and 3.6 s, the pitch angles (from highest to lowest) follow the order 1 > 3 > 2 > 4 and 1 > 3 > 4 > 2, respectively. At t = 3.4 s and 3.9 s, the order changes to 4 > 2 > 1 > 3. Compared to the fixed-torso configuration (curve 1), curves 2, 3, and 4 show reduced pitch angles during the first peak. In the second peak, curves 2 and 4 exhibit slightly larger amplitudes than curve 1, while curve 3 consistently maintains lower pitch values throughout. These results indicate that under torso yawing motion, the center-of-mass-based trajectory provides the most favorable coordination performance for the bionic-torso quadruped robot during trot gait locomotion.

Figure 22d presents a comparison of the robot’s roll angle under different torso yaw trajectories and the fixed-torso condition, while Figure 22e provides a magnified view of the roll angle variation during the interval from 3 to 4 s. As shown in Figure 22d, all four curves demonstrate periodic fluctuations after t = 2 s. However, notable differences are observed in both phase and amplitude. Among them, curve 3 exhibits the smallest roll angle variation, whereas curve 1 presents the largest fluctuation. From Figure 22e, it can be further observed that curves 1 and 4 follow a similar trend, with curve 4 having a slightly smaller amplitude. The torso yaw motion in trajectory 4 contributes little to the reduction in body roll. Curve 2 shows a certain degree of phase delay, and its roll fluctuation is slightly greater than that of curve 3. Notably, curve 3 maintains the most stable roll angle, consistently around −0.3°. During the 3–4 s interval, the differences between the maximum and minimum roll angles for curves 1 through 4 are 2.0527°, 1.5273°, 0.9719°, and 1.8437°, respectively. These results indicate that torso yawing motion is effective in reducing body roll during trot gait. In particular, the center-of-mass-based trajectory (curve 3) achieves the best performance, significantly enhancing the stability of the bionic-torso quadruped robot.

The comparison of maximum joint torques under fixed-torso and yawing-torso conditions is shown in Figure 23. Figure 23a presents the peak torques of the hip and knee joints of each leg, where FR, FL, HL, and HR denote the front-right, front-left, hind-left, and hind-right legs, respectively. The labels R1, W2, W3, and W4 correspond to the same torso motion strategies, as defined in Figure 23b.

As shown in Figure 23a, the hip joint torques of the left foreleg and right hind leg are approximately equal, both around 80 N·m, while those of the right foreleg and left hind leg are also similar, approximately 60 N·m. For the knee joints, the torque of the right hind leg is slightly lower than that of the left foreleg but still higher than those of the right foreleg and left hind leg. This pattern is consistent with that observed in the hip joint torques, which reflects the alternating leg movement characteristic of the diagonal trot gait.

Figure 23b presents the maximum joint torques of the actuators in the bionic parallel torso. Joint 1 to Joint 6 correspond to the six actuated joints of the torso mechanism. Due to the structural asymmetry between the front and rear body segments of the quadruped robot, the robot exhibits left–right asymmetry during diagonal gait locomotion. As a result, the joint torques at the torso actuators do not follow a clear symmetric distribution.

## 6. Results

A comparative simulation study was conducted to evaluate the locomotion performance of a quadruped robot equipped with a bionic parallel-actuated torso versus a fixed-torso configuration under a trot gait at a target velocity of 0.3 m/s. Kinematic results showed that the active torso configuration achieved improved locomotion efficiency, attaining an average forward velocity of 0.286 m/s—corresponding to a 4.6% deviation from the desired speed—whereas the fixed torso configuration reached 0.275 m/s, with an 8.3% deviation. In addition, the actively modulated torso exhibited enhanced lateral stability, as reflected by a 31.65% reduction in *CoM* displacement along the Y-axis, while maintaining comparable vertical *CoM* oscillations along the *Z*-axis.

Postural dynamics further demonstrated the advantages of active torso modulation. The *CoM*-based yaw control strategy effectively reduced body roll amplitude and preserved bounded attitude stability, despite greater variations in the pitch axis induced by periodic torso movements. These improvements became evident following approximately 3 s of gait stabilization, after which the robot’s attitude angles converged to regular, periodic patterns. Dynamic performance metrics also highlighted the benefits of the actively actuated torso. Ground reaction force (*GRF*) analysis revealed a 4.1% reduction in peak vertical loading at the foot ends—278 N compared to 290 N—indicating smoother ground contact and diminished impact forces. Joint torque analysis demonstrated more balanced distribution of actuation efforts across the limb joints and the six actuated degrees of freedom of the torso, supporting improved mechanical coordination during locomotion. Furthermore, foot-end workspace analysis based on actual joint trajectories revealed that the *CoM*-based torso control strategy expanded the reachable area by approximately 54% relative to the fixed-torso configuration. This increased workspace enhances foot placement flexibility and overall adaptability during dynamic gait execution.

Collectively, these results confirm that integrating an actively modulated bionic parallel torso improves trot gait performance in terms of stability, impact attenuation, and maneuverability. Among the tested control strategies, the *CoM*-based trajectory approach proved particularly effective in reducing lateral deviations and postural fluctuations, offering a robust solution for coordinated whole-body locomotion in quadruped robots.

## 7. Discussions and Conclusions

This study proposes a quadruped robot equipped with a six-degree-of-freedom (6-*DOF*) bionic parallel-actuated torso, inspired by the compliant and flexible spinal structures found in quadrupedal animals. A complete kinematic model of both the torso and the full-body system is established, and a model predictive control (*MPC*) strategy based on a variable center of mass (*CoM*) is introduced to address the inertia variations induced by torso motion, thereby enabling real-time coordination between the torso and the legs. Simulation experiments conducted under a trot gait at a target speed of 0.3 m/s demonstrate that the active torso configuration significantly improves locomotion performance. Specifically, it enhances *CoM* stability, reduces joint torques, expands the reachable foot workspace, mitigates impact forces, and increases forward velocity. These findings validate the effectiveness of active torso–leg coordination in enhancing dynamic stability and maneuverability.

Although various bionic torso structures and control strategies have been proposed [7,8,9,10,11,12,13,14,15,16,17,18,19,20], most are still limited by low degrees of freedom, constrained motion in the sagittal plane, or simplified structures intended for lightweight platforms. In contrast, the proposed 6-DOF parallel mechanism features symmetrical actuation, bidirectional compliance (including extension and compression), and multi-axis motion capability, enabling a broader range of posture regulation and dynamic whole-body control. In our previous studies [21,22,23], we systematically compared this structure with traditional single-DOF and passive designs and confirmed its advantages in motion diversity, impact attenuation, and torso–limb coordination. However, while offering improved flexibility and coordination, this design also introduces new challenges and limitations. In particular, some individual limbs within the current parallel structure suffer from insufficient strength and may face fatigue failure under prolonged multidirectional loading. Additionally, compared with serial architectures, actuators in the parallel mechanism are more susceptible to axial impacts caused by rapid torso postural changes, especially during high-dynamic tasks such as running or jumping. Future study must therefore focus on optimizing structural layouts and material strength to enhance the shock resistance and dynamic load-bearing capacity of the parallel mechanism without compromising its motion performance.

Furthermore, we acknowledge that, compared with low-*DOF* alternatives, the proposed system entails higher mechanical complexity, actuator performance requirements, and control precision demands. Unlike commercially available platforms such as MIT Cheetah (Massachusetts Institute of Technology, Cambridge, United States), ANYmal (ANYbotics, Zurich, Switzerland), and Laikago (Unitree Robotics, Hangzhou, China)—which utilize rigid or low-*DOF* torso designs—our approach prioritizes motion compliance and postural adaptability, making it more suitable for research- and exploration-oriented scenarios requiring high mobility and dynamic versatility rather than immediate industrial deployment. To further improve the system’s practical applicability, future research will explore diverse torso motion patterns under different gait modes, develop hardware prototypes, and conduct experimental validation against current state-of-the-art quadruped robot platforms.

## Figures and Tables

**Figure 1 biomimetics-10-00335-f001:**
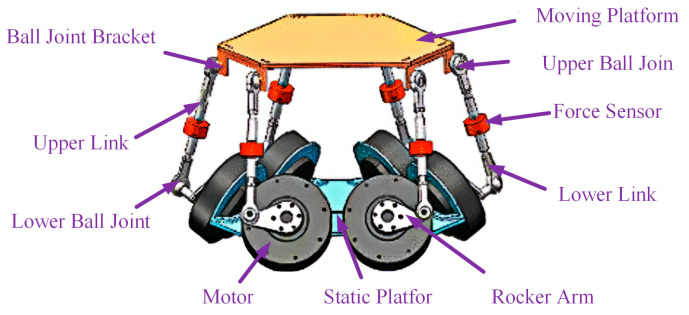
The structure of the bionic active torso.

**Figure 2 biomimetics-10-00335-f002:**
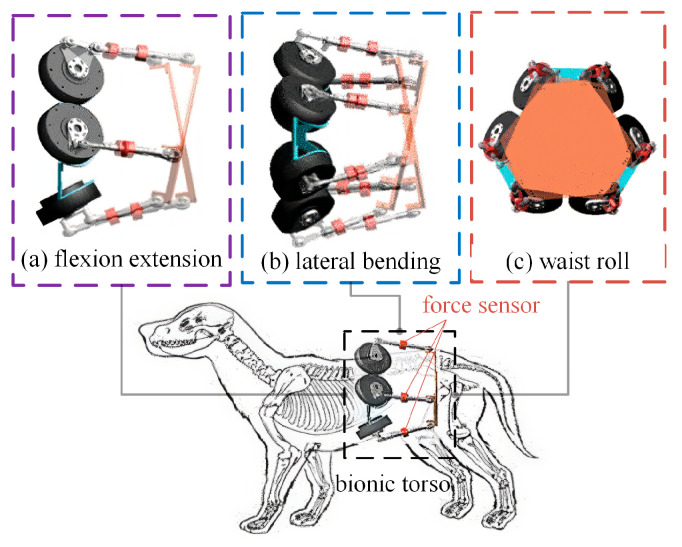
Motion patterns of the torso. (**a**) flexion-extension pattern of the torso; (**b**) later bending pattern of the torso; (**c**) waist roll pattern of the torso.

**Figure 3 biomimetics-10-00335-f003:**
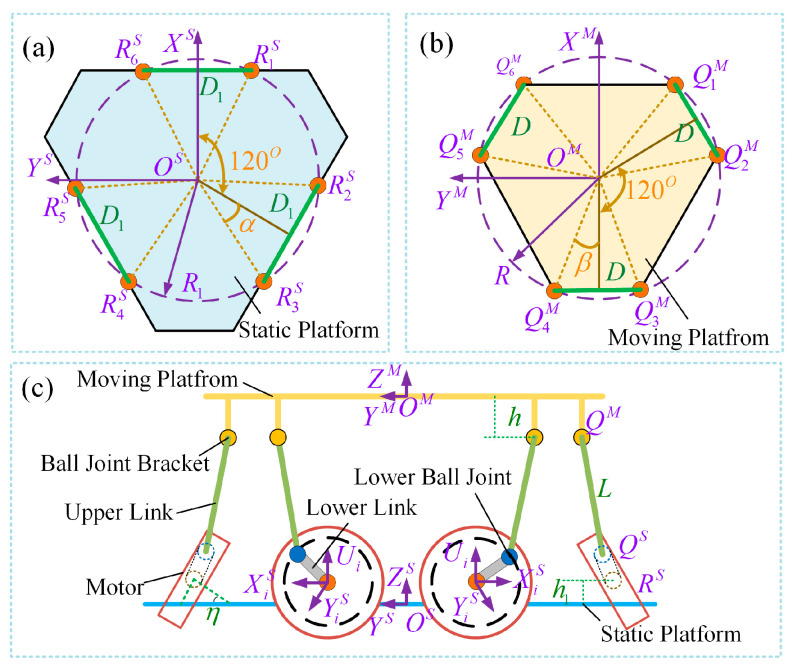
(**a**) Geometric configuration of the static platform; (**b**) Geometric configuration of the moving platform. The orange points represent the installation positions of the torso motor; (**c**) Geometric configuration of the active torso.

**Figure 4 biomimetics-10-00335-f004:**
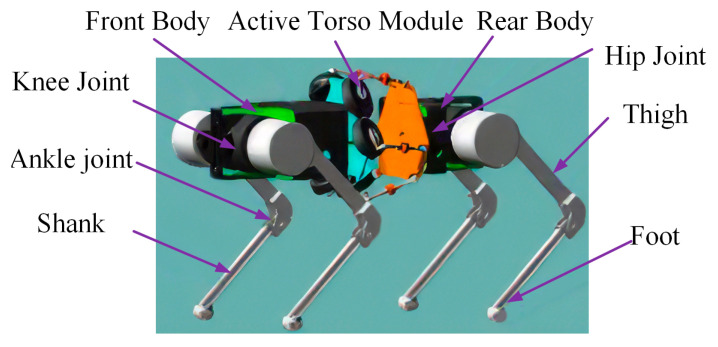
Structure of the bionic active torso quadruped robot.

**Figure 5 biomimetics-10-00335-f005:**
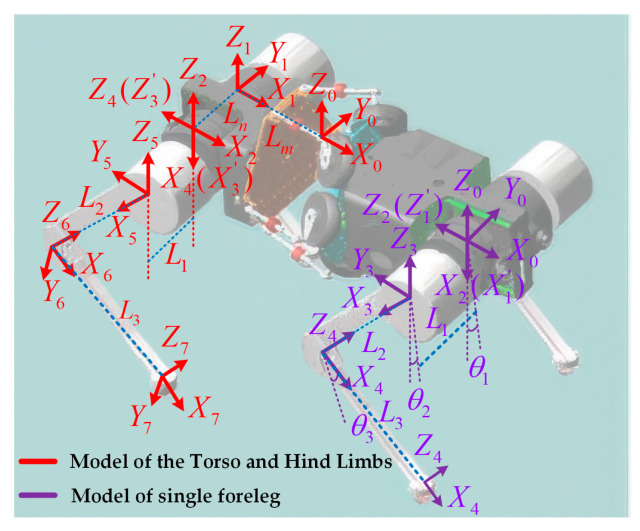
D-H coordinate frames of the bionic quadruped robot with active torso.

**Figure 6 biomimetics-10-00335-f006:**
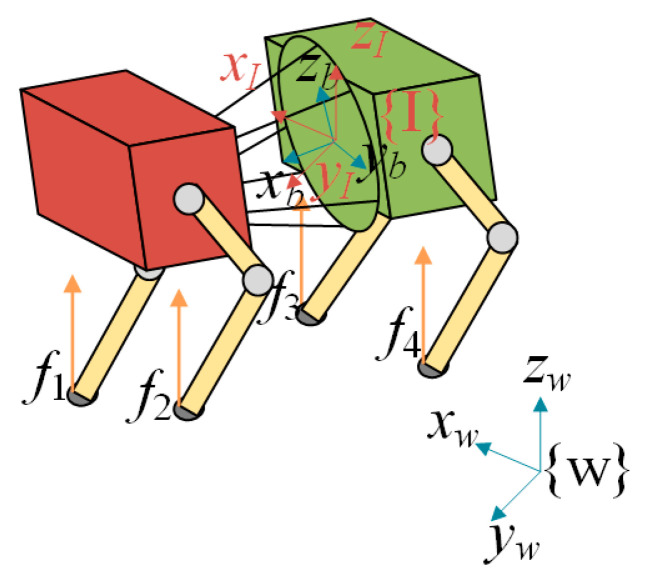
Single rigid body simplified model.

**Figure 7 biomimetics-10-00335-f007:**
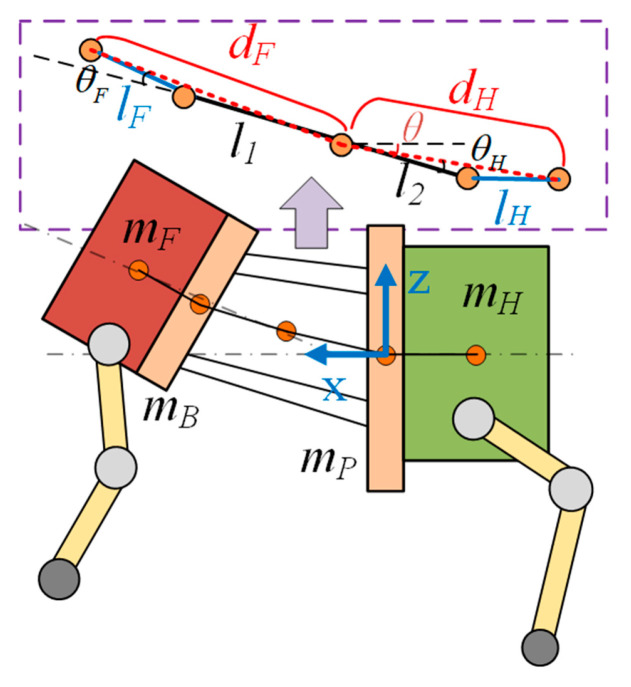
Simplified model for torso inertia calculation of a bionic quadruped robot. The orange points represent the *COM* of the robot’s components. From left to right, they correspond to the *COM* of the front body, static platform, torso, moving platform, and rear body.

**Figure 8 biomimetics-10-00335-f008:**
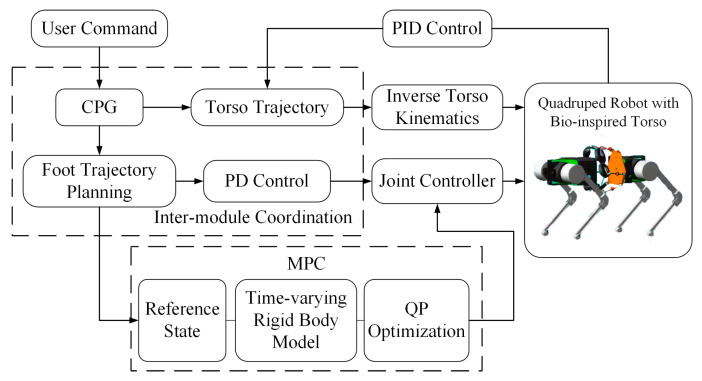
Overall control framework of the quadruped robot with a bionic parallel torso.

**Figure 9 biomimetics-10-00335-f009:**
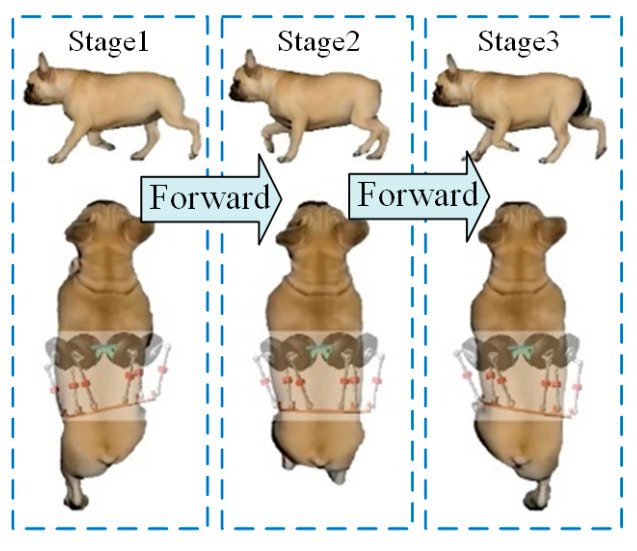
Illustration of diagonal trotting motion in quadrupedal canines. The blue arrows represent the forward motion of the dog, while the blue box indicates the movement state of the dog’s torso and legs during stages 1–3.

**Figure 10 biomimetics-10-00335-f010:**
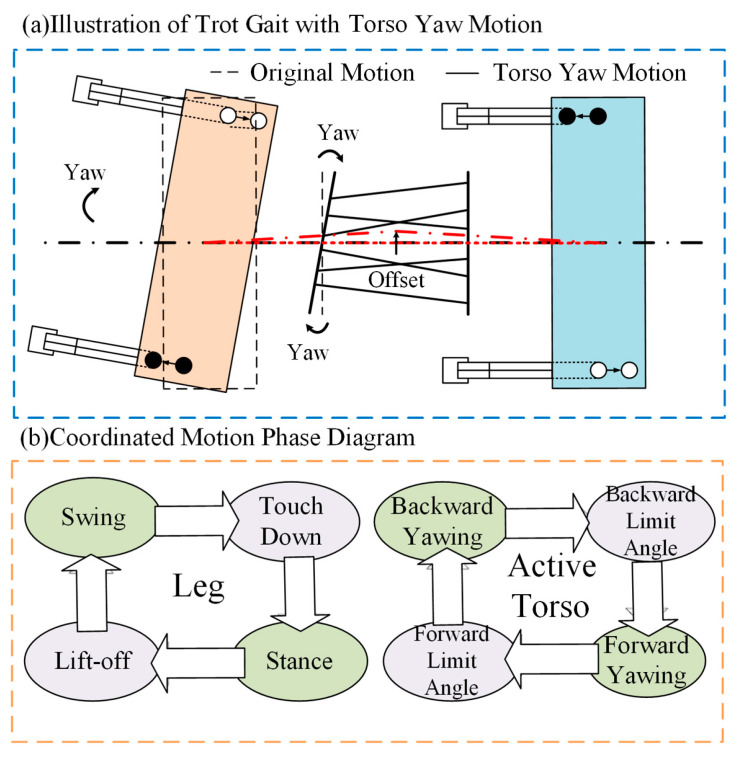
Illustration of coordinated motion between torso and legs. (**a**) Illustration of Tort Gait with Torso Yaw Motion. The black dashed line represents the original states of the body and torso, the red dashed line represents the longitudinal axis of the torso, and the black arrow indicates the yaw motion of the torso. (**b**) Coordinated Motion Phase Diagram.

**Figure 11 biomimetics-10-00335-f011:**
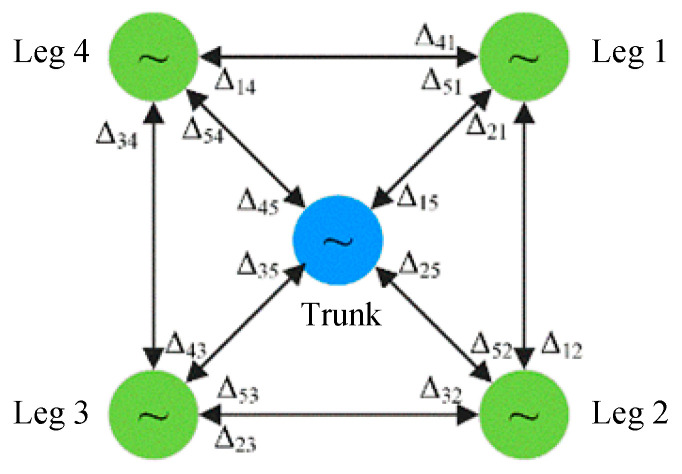
*CPG* network structure.

**Figure 12 biomimetics-10-00335-f012:**
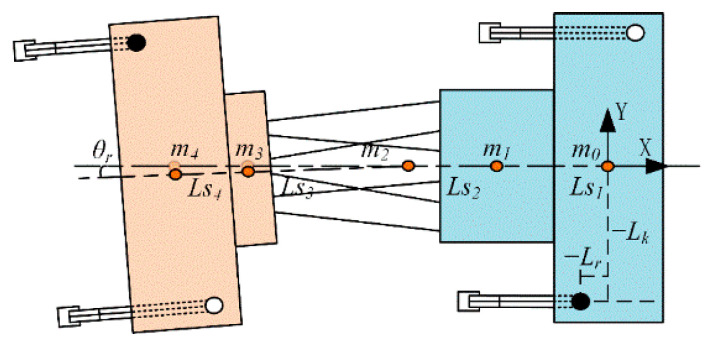
Illustration of the *CoM* during torso yaw deflection. The orange points represent the *COM* of the robot’s components. From left to right, they correspond to the *COM* of the rear body, moving platform, torso, static platform, and front body.

**Figure 13 biomimetics-10-00335-f013:**
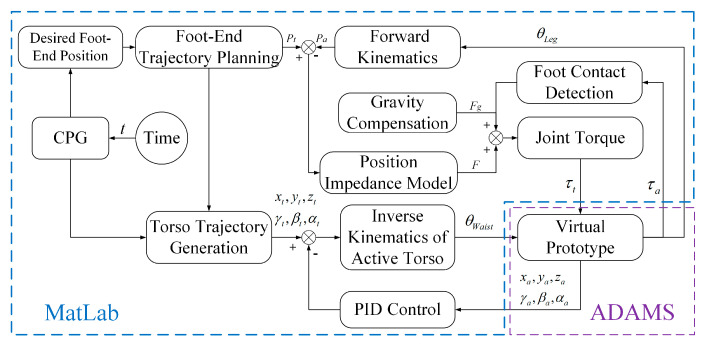
Algorithm verification framework for bionic active torso quadruped robot.

**Figure 14 biomimetics-10-00335-f014:**
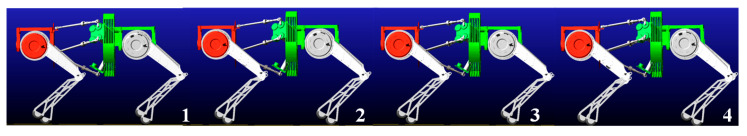
Quadruped robot performing a trot gait with a fixed torso, the red area represents the front body, and the green area represents the fixed torso.

**Figure 15 biomimetics-10-00335-f015:**
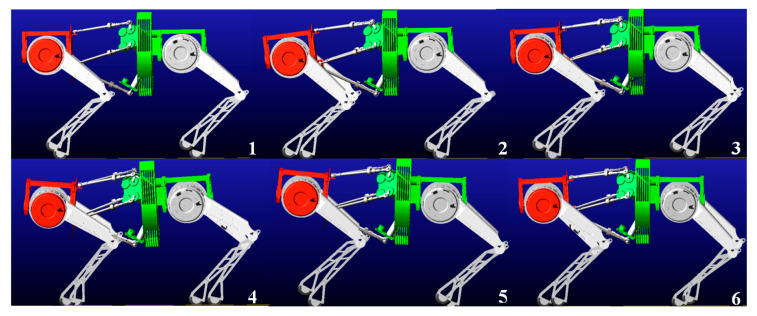
Quadruped robot performing a trot gait with a bionic active torso, the red area represents the front body, and the green area represents the active torso.

**Figure 16 biomimetics-10-00335-f016:**
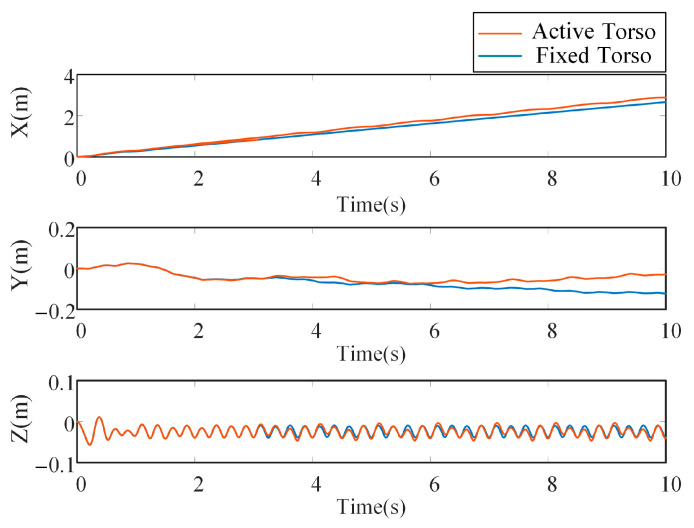
Comparison of the *CoM* displacement trajectories with different torsos.

**Figure 17 biomimetics-10-00335-f017:**
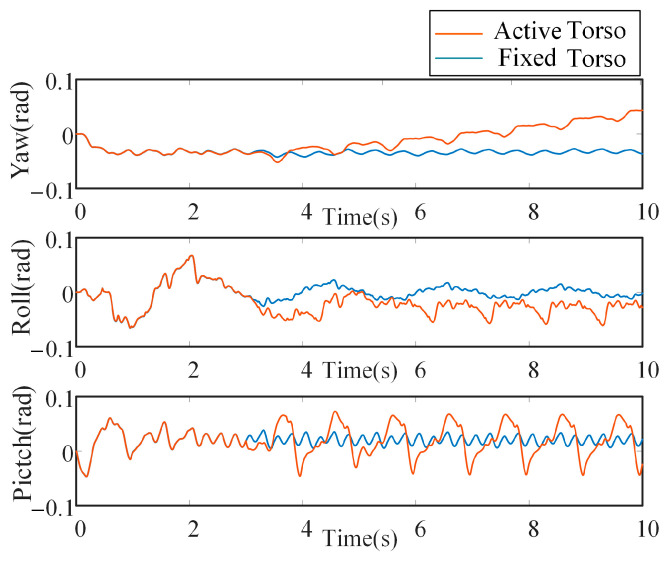
Comparison of attitude angle with different torsos.

**Figure 18 biomimetics-10-00335-f018:**
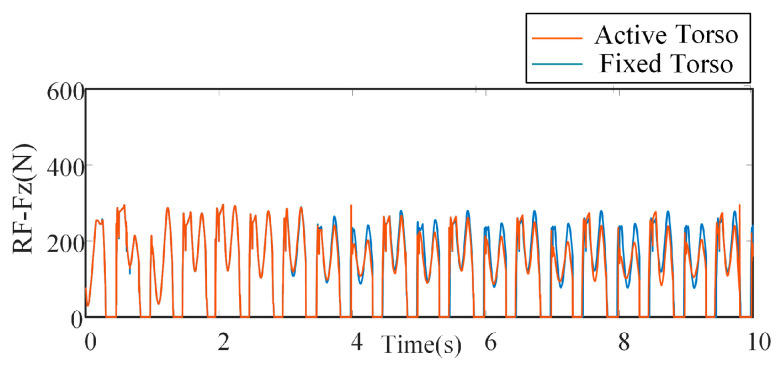
Comparison of foot-end vertical ground reaction forces with different torsos.

**Figure 19 biomimetics-10-00335-f019:**
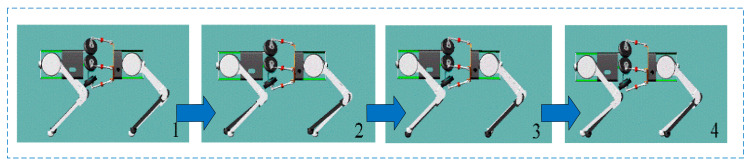
Trot gait locomotion of the quadruped robot with a rigid torso. Frame 1 represents the right front leg and left rear leg in the swinging phase. Frame 2 represents the right front leg and left rear leg continuing to swing forward. Frame 3 represents the left front leg and right rear leg beginning the swinging phase. Frame 4 represents the left front leg and right rear leg continuing to swing forward. The blue arrows indicate the robot’s forward movement.

**Figure 20 biomimetics-10-00335-f020:**
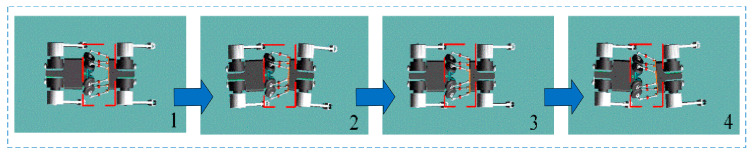
Coordinated locomotion of the quadruped robot with an active bionic parallel torso. The red dashed box indicates the bionic torso movement. Frame 1 represents the front and rear bodies being parallel along the horizontal axis. Frame 2 represents the torso following the swinging of the legs and yaw to the right. Frame 3 represents the front and rear bodies being parallel along the horizontal axis. Frame 4 represents the torso following the swinging of the legs and yaw to the left. The blue arrows indicate the robot’s forward movement.

**Figure 21 biomimetics-10-00335-f021:**
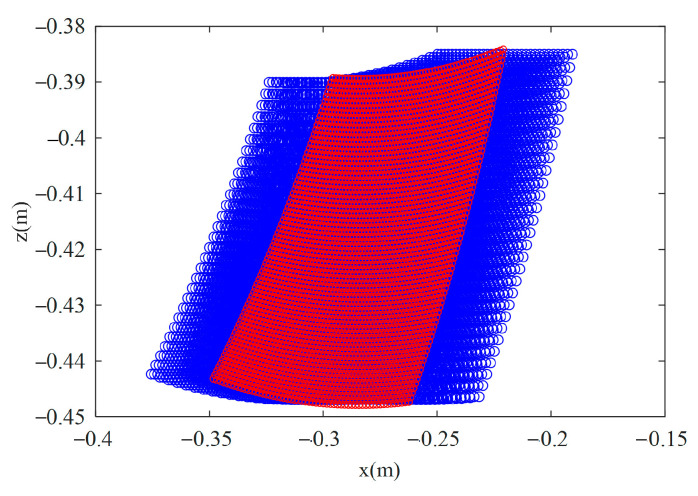
Workspace of the foot end. The red area represents the workspace of the foot-end when the torso is fixed, and the blue area represents the foot-end workspace when the torso undergoes a yaw of 7.39° based on the *COM* trajectory.

**Figure 22 biomimetics-10-00335-f022:**
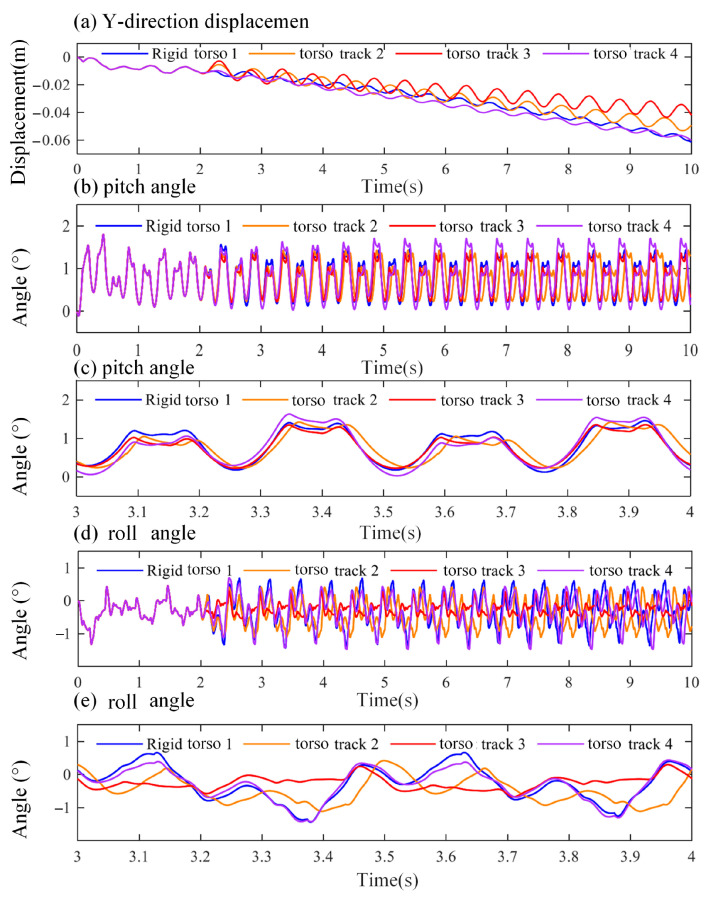
Comparison of robot pose with different structures and control strategies. (**a**) Displacement of the robot in the Y-direction under different structures and control strategies; (**b**) Pitch angle of the robot under different structures and control strategies; (**c**) Pitch angle of the robot between 3–4 s under different structures and control strategies; (**d**) Roll angle of the robot under different structures and control strategies; (**e**) Roll angle of the robot between 3–4 s under different structures and control strategies.

**Figure 23 biomimetics-10-00335-f023:**
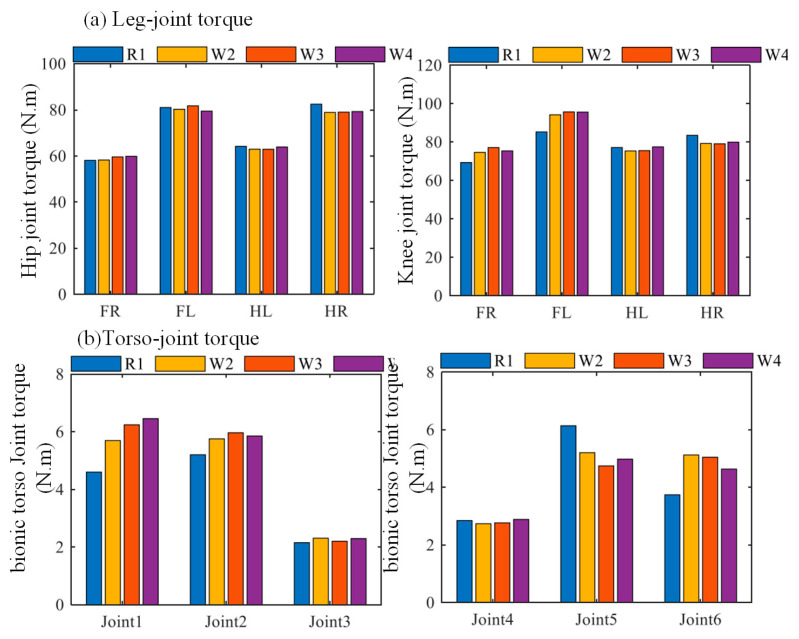
Comparison of maximum joint torques of the robot. (**a**) Maximum torque of the leg’s hip joint and knee joint motors under different structures and control strategies. (**b**) Maximum torque of the torso motor under different structures and control strategies,. R1, W2, W3, and W4 correspond to Rigid Torso 1 (fixed torso), Torso Track 2 (CPG-based trajectory), Torso Track 3 (CoM-based trajectory), and Torso Track 4 (ZMP-based trajectory), respectively.

**Table 1 biomimetics-10-00335-t001:** Structural parameters of the active torso.

Symbol	Structural Parameter	Value
η	Motor inclination angle	π/6 rad
R	Radius of the $Q^{M}$	0.136 m
$R_{1}$	Radius of the $R^{S}$	0.132 m
D	Distance between $Q^{M}$	0.110 m
$D_{1}$	Distance between $R^{S}$	0.088 m
r	Rocker length	0.040 m
L	Link length	0.140 m
h	Height of $Q^{M}$ from the moving platform	0.015 m
$h_{1}$	Height of $R^{S}$ from the static platform	0.0235 m

**Table 2 biomimetics-10-00335-t002:** Major parameters of the bionic active torso quadruped robot.

Symbol	Description	Value
$L_{m}$	Distance from the torso *CoM* to the front/rear hip joint	0.286 m
$L_{n}$	Distance from the longitudinal midline of The body to the hip joint	0.058 m
$L_{1}$	Distance between the Hip joint and the knee joint	0.155 m
$L_{2}$	Length of the thigh	0.323 m
$L_{3}$	Length of the shank	0.304 m
H	Vertical distance from the *CoM* to the foot end	0.450 m

**Table 3 biomimetics-10-00335-t003:** DH Parameters of a single leg.

*i*	*α*°	*a* (m)	*θ*°	*d* (m)
2	0	0	*θ* _1_	0
3	90°	0	*θ* _2_	−*L*_1_
4	0	*L* _2_	*θ* _3_	0
5	0	*L* _3_	0	0

## Data Availability

Data are contained within the article.

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
