# Peer review of "Coordinated Locomotion Control for a Quadruped Robot with Bionic Parallel Torso"

_biomimetics, 2025, doi:10.3390/biomimetics10050335_

Round 1

Reviewer 1 Report

Comments and Suggestions for Authors
  1. I suggest to use more specific keywords, these are too holistic
  2. I suggest to use newer references for your article. 2023 ، 2024 articles not covered
  3. I suggest to use more clear and high quality figures. figures 1,2,3&5 are not clear and need to be replaced
  4. result section must be quantitative and compare result with other  

Reviewer 2 Report

Comments and Suggestions for Authors

The research is focused on a developement and modeling of the quadruped robot equipped with 6-DOF parallel-actuated torso inspired by a quadrupedal mammals motion patterns. Authors derived a 6-DOF structure for a parallel-actuated torso and developed its kinematic model and the whole-body kinematic robot of the robot. The advantage of the proposed contruction is the ability to make the rigid torso simulation on the same model. To control the robot motion authors provided an MPC-based discrete control algorithm witha variable center of masses consideration. The comparative simulations demonstrated the inhancements of the gait parameters such as postural stability and locomotion efficiency of the quadruped robot with developed 6-DOF torso against similar robots with rigid torso.

Reviewer 3 Report

Comments and Suggestions for Authors

Comments to Authors:

The recommendation of this reviewer is “Accept after minor revisions”, the following suggestions are for the authors’ reference.

(1) The MPC strategy is introduced but not thoroughly analyzed. The choice of cost function, constraints, and tuning parameters is not well-justified. A deeper discussion of these aspects would improve the clarity and robustness of the control design.

(2) The figures in the manuscript are not clear enough, please improve the clarity of the figures.

(3) The paper does not adequately address the limitations of the proposed design. For example, the energy efficiency of the 6-DOF parallel torso is not discussed, nor are the potential trade-offs between flexibility and stability. Please provide a discussion of the limitations of the proposed design, such as energy efficiency, scalability, or potential trade-offs between flexibility and stability.

(4) While the paper demonstrates improved performance in simulations, it does not clearly articulate the practical relevance of the proposed design. For example, how does the 6-DOF parallel torso compare to existing solutions in terms of cost, complexity, or scalability?

The lack of a clear comparison with state-of-the-art quadruped robots limits the paper's impact. The authors should provide a more comprehensive review of existing biomimetic quadruped designs and control strategies to situate their work within the broader context of the field.

Reviewer 4 Report

Comments and Suggestions for Authors

The paper presents the design, analysis and simulations of a 6-DOF torso system for a quadruped robots
The paper is well written, comprehensible and offer an original contribution to biomimetics
However the discussion of results could be enhanced to emphatises the differences with previous researches,

Few recommendations are addressed to authors:
- please check the terms used: better force sensor or load cell ?
- please use the same font size in equations e.g. 29 and 32 are different in size and symbols (missing dot in 32)
- L332 for eq.nr.35 is different from the others
- suggest to rename section 6. as Conclusions (lines 709-718 look like an abstract in the wrong place)
- it should be interesting to see a comparison of author's torso with some of the solutions mentoned in introduction, [7-15]

Round 2

Reviewer 1 Report

Comments and Suggestions for Authors
  1. keywords are not acceptable, I suggest use better keywords that are shorter and more specific
